# Quality control of transcription start site selection by nonsense-mediated-mRNA decay

**Christophe Malabat[1†], Frank Feuerbach[1†], Laurence Ma[2], Cosmin Saveanu[1], Alain Jacquier[1]***

[1]Institut Pasteur, UMR3525, Génétique des Interactions Macromoléculaires, Centre National de la Recherche Scientifique, Paris, France; [2]Plate-Forme Génomique, Institut Pasteur, Paris, France

**Abstract** Nonsense-mediated mRNA decay (NMD) is a translation-dependent RNA quality-control pathway targeting transcripts such as messenger RNAs harboring premature stop-codons or short upstream open reading frame (uORFs). Our transcription start sites (TSSs) analysis of *Saccharomyces cerevisiae* cells deficient for RNA degradation pathways revealed that about half of the pervasive transcripts are degraded by NMD, which provides a fail-safe mechanism to remove spurious transcripts that escaped degradation in the nucleus. Moreover, we found that the low specificity of RNA polymerase II TSSs selection generates, for 47% of the expressed genes, NMD-sensitive transcript isoforms carrying uORFs or starting downstream of the ATG START codon. Despite the low abundance of this last category of isoforms, their presence seems to constrain genomic sequences, as suggested by the significant bias against in-frame ATGs specifically found at the beginning of the corresponding genes and reflected by a depletion of methionines in the N-terminus of the encoded proteins.

*For correspondence: alain.jacquier@pasteur.fr

†These authors contributed equally to this work

**Competing interests:** The authors declare that no competing interests exist.

## Introduction

Recent advances in sequencing technologies led to the detection of a wealth of new RNA transcripts and revealed that eukaryotic genomes are pervasively transcribed. In human cells, roughly 75% of the genome gives rise to RNA transcripts of various length but only an estimated 2% corresponds to protein-coding messenger RNAs (mRNAs) (*Djebali et al., 2012*). Even in a compact genome such as the budding yeast *Saccharomyces cerevisiae* (*S. cerevisiae*), hundreds of such pervasive non-coding RNAs (ncRNAs) were identified in addition to the stable ncRNAs such as transfer RNAs (tRNAs), small nucleolar RNAs (sn(o)RNAs), and ribosomal RNA (rRNAs) (*Neil et al., 2009*; *Xu et al., 2009*; *van Dijk et al., 2011*; *Geisler et al., 2012*).

Yeast ncRNAs are transcribed from nucleosome-free regions (NFRs) present throughout the genome, most often at gene promoters and terminators, and part of these RNAs represent by-products of divergent transcription initiation (*Neil et al., 2009*; *Xu et al., 2009*). Different mechanisms of RNA quality control prevent their accumulation in wild-type cells. In the nucleus, transcription from bidirectional promoters is restricted by early termination-coupled RNA degradation pathways (*Arigo et al., 2006*; *Thiebaut et al., 2006*; *Almada et al., 2013*; *Ntini et al., 2013*). In the yeast *S. cerevisiae*, degradation-coupled transcription termination of pervasive transcripts relies on the Nrd1-Nab3-Sen1 (NNS) complex, which interacts with the carboxy-terminal domain (CTD) of the RNA polymerase II (RNAPII) phosphorylated on serine 5 and triggers termination upon recognition of short sequences on the nascent RNA (*Arigo et al., 2006*; *Thiebaut et al., 2006*; *Gudipati et al., 2008*; *Vasiljeva et al., 2008*; *Schulz et al., 2013*). Early termination of these transcripts (named CUTs for

**eLife digest** Eukaryotes such as animals, plants and fungi store their DNA within the nucleus of each of their cells. Genes within this DNA contain the instructions needed to make molecules of RNA; some of which can leave the nucleus and be decoded to build proteins. However, not all of the DNA that is copied into RNA actually codes for proteins. Instead, some RNA molecules are important parts of the cell's protein-making machinery in their own right, and others help to regulate the expression of genes as RNAs or proteins.

Nevertheless, many non-coding RNAs don't have such clear roles. Often these RNAs—which are called 'pervasive transcripts'—are quickly destroyed within the nucleus, but it is likely that some molecules will escape this quality-control mechanism. If the cell's protein-making machinery decodes these RNAs, it could lead to the production of faulty or harmful proteins. Recent research suggested that another quality-control mechanism, which typically eradicates incorrectly processed protein-coding RNAs, could also destroy unneeded or harmful pervasive transcripts. But it was not clear how common it was for this process—called 'nonsense-mediated decay'—to be used for this purpose.

Now Malabat, Feuerbach et al. have engineered yeast cells that lacked either the genes required to carry out nonsense-mediated decay or the ability to destroy RNA molecules in the nucleus. Experiments with these yeast cells revealed that about half of all pervasive transcripts can be destroyed via nonsense-mediated decay; this suggests that this mechanism serves as a fail-safe to prevent the build-up of these potentially harmful molecules.

Malabat, Feuerbach et al. also revealed that the enzyme complex that copies gene sequences to make RNA molecules will often also copy some extra DNA sequence from before the start of the gene. On the other hand, it is also common for this enzyme complex to miss the start of the gene and produce an RNA molecule that lacks some of the instructions needed to build the correct protein. Further experiments showed that in yeast these two kinds of incorrectly made protein-coding RNAs could both be identified and destroyed by nonsense-mediated decay as well. The next challenge will be to see to what extent these phenomena are conserved in other eukaryotes.

cryptic unstable transcripts) is coupled with the recruitment of the Trf4-Air2-Mtr4 (TRAMP) complex and the Rrp6-containing nuclear RNA-exosome for rapid degradation (*Wyers et al., 2005*; *Davis and Ares, 2006*; *Tudek et al., 2014*). However, a significant proportion of the pervasive transcripts can escape this early nuclear quality-control step and be exported to the cytoplasm where they are targeted for degradation by the cytoplasmic 5′-3′ exonuclease Xrn1 (XUT; *van Dijk et al., 2011*). Some pervasive transcripts seem to be immune enough to both surveillance pathways to be detectable in wild-type cells and are called SUTs (stable unannotated transcripts; *Xu et al., 2009*). The distinction between these different classes of transcripts is not very stringent. Their behavior can be similar, as illustrated by the fact that in the absence of Xrn1p the average levels of SUTs and CUTs increase by 7.9-fold and 3.6-fold respectively (*van Dijk et al., 2011*). What makes pervasive transcripts (XUTs in the first instance) highly sensitive to Xrn1p-dependent cytoplasmic degradation remains to be determined but may be linked to the presence of small spurious open reading frames (ORF) in these transcripts that will make them substrates for the nonsense-mediated mRNA decay (NMD) as has been recently demonstrated for a new class of unannotated transcripts (*Smith et al., 2014*).

Originally described more than two decades ago as a quality-control pathway targeting for degradation mRNAs containing a premature termination codon (*Leeds et al., 1991*), NMD is a translation-coupled quality-control pathway affecting cytoplasmic transcripts with restricted coding capacities (for a review see *Kervestin and Jacobson, 2012*). NMD targets include mRNAs harboring 'upstream open reading frames' (uORFs) (*He et al., 2003*; *Arribere and Gilbert, 2013*) and unspliced or incorrectly spliced transcripts exported to the cytoplasm (*Sayani et al., 2008*; *Kawashima et al., 2014*) accounting for roughly 10–15% of mRNAs in yeast as well as in human cells (*Kervestin and Jacobson, 2012* and references therein). This number may be an underestimate since recent studies underscored the structural heterogeneity of most mRNAs at both their 5′ and 3′ -ends (*Ozsolak et al., 2010*; *Arribere and Gilbert, 2013*; *Pelechano et al., 2013*; *Waern and Snyder, 2013*).

Transcript heterogeneity at the 5′-end is particularly relevant for NMD targeting since the use of different transcription start sites (TSSs) will generate, for a given gene, mRNAs with different 5′-UTRs

that may include uORFs. Although NMD targets were previously analysed genome-wide using NMD-deficient strains of *S. cerevisiae*, the techniques used to perform these analyses (*He et al., 2003*; *Kawashima et al., 2014*; *Smith et al., 2014*) did not allow monitoring such individual transcript isoforms and were thus susceptible to have missed a large number of transcripts degraded by this pathway. In order to address this question in a global and systematic manner, we used a modified 5′-RACE approach (*Hashimoto et al., 2009*; *Arribere and Gilbert, 2013*) to perform a genome-wide analysis of TSSs in wild-type *S. cerevisiae* cells as well as in cells deficient for nuclear and NMD RNA quality-control pathways. Minor transcript isoforms targeted for degradation by NMD were identified for almost half of yeast protein-coding genes, underscoring the low specificity of TSS selection by RNAPII. In particular, our study revealed for the majority of protein-coding genes the use of TSSs downstream the ATG start codons. This phenomenon has the potential to generate N-terminally truncated proteins if the first ATG encountered by ribosomes translating this 5′ truncated transcript is in the right reading frame. Such an undesirable outcome seems, however, to be counteracted by the significant depletion of in-frame relative to out-of-frame ATGs at the beginning of protein-coding genes in yeast. This bias increases the probability of such transcripts to have very short coding regions and be efficiently degraded by NMD.

Our analysis also showed that NMD restricts the accumulation of cryptic transcripts initiating inside transcribed ORFs as a consequence of altered chromatin structure and provides a fail-safe control mechanism for the removal of pervasive transcripts that escaped degradation by the nuclear quality-control pathway, reminiscent of what has been previously shown for some unspliced pre-mRNAs (*Sayani and Chanfreau, 2012*). This includes not only a large fraction of the XUTs and SUTs but also a fraction of the CUTs and previously unannotated transcripts, which accumulate to substantial levels only when both pathways are inactive.

## Results

### TSS sequencing technique

To identify TSSs genome-wide and with high specificity we used a modified genomic 5′-RACE approach (*Hashimoto et al., 2009*; *Arribere and Gilbert, 2013*) that we called TSS sequencing, which involves a biotin purification step and allows the selective enrichment of the 5′-ends of capped transcripts (see *Figure 1A* and 'Materials and methods'). We evaluated the 'false-discovery' rate of the method for the identification of TSSs (that is, the proportion of sequencing reads not actually mapping to the 5′-end of capped RNAs) by comparing libraries made using samples treated or not with the tobacco acid pyrophosphatase (TAP), which is required for efficient and specific ligation of the biotinylated primer to the 5′-end of capped-RNAs. To extend this analysis to transcripts that are unstable in wild-type cells, we prepared libraries using RNAs extracted from *upf1Δrrp6Δ* double mutant strains. To minimize polymerase chain reaction (PCR) amplification biases and provide an internal control, the *S. cerevisiae* poly(A)$^+$ RNAs, treated or not with TAP, were mixed with an equal amount of TAP-treated poly(A)$^+$ RNAs from *Schizosaccharomyces pombe* just prior to the ligation step. After normalization using the *S. pombe* sequencing reads and removal of the ribosomal DNA ones, these experiments generated 2,844,877 reads when TAP was used compared with 72,435 reads when TAP was omitted. The ratio between these two numbers provides an upper limit for the false-discovery rate of TSS identification of 2.5%. Analysis of the cumulative 5′-end read counts per nucleotide around the start codon for all protein-coding genes showed that, genome-wide, 77% (2,190,211) of the reads obtained for TAP-treated samples could be mapped within a 200 nucleotide region upstream of the start codons with a maximum at around 30 nucleotides upstream from ATGs, while only 22% mapped to the same region when this treatment was omitted (*Figure 1B*). However, even in the latter case, the number of mapped reads also peaked at around 30 nucleotides upstream of the ATGs, suggesting that a substantial proportion of these sequences likely correspond to genuine TSSs even though they were generated in the absence of TAP treatment. These data suggest that the real false-discovery rate for TSS is substantially less than 2.5%.

### Repeatability and reproducibility of TSS sequencing

To estimate the repeatability of the experiment, two independent biological replicates were generated in parallel from wild-type cells and sequenced on different lanes of an Illumina HiSeq 2500 sequencer. The number of reads mapping to the same genomic position were highly correlated

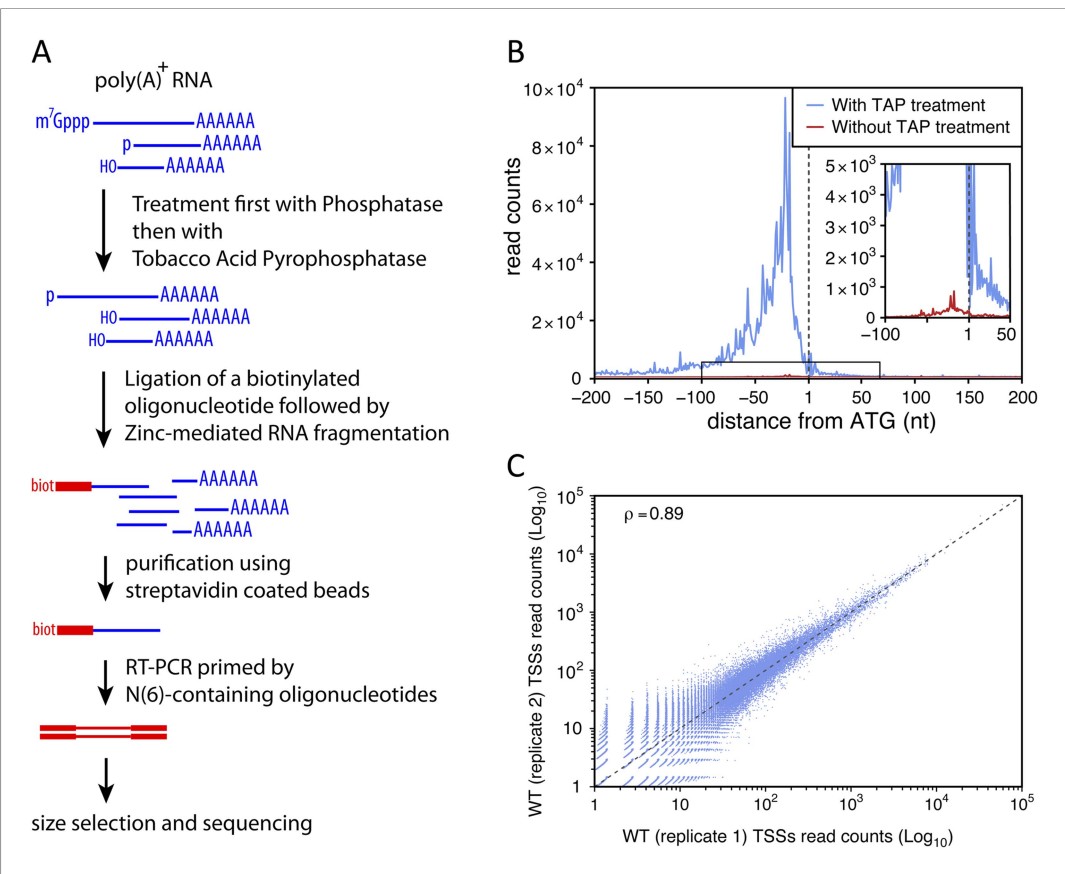

**Figure 1.** Transcription start site sequencing. (**A**) Schematic view of the methodology used to produce the transcription start site (TSS) sequence-tag libraries (RNA molecules are in blue, DNA molecules in red). (**B**) All protein-coding genes were aligned on the A of their annotated ATG start codon and the distribution of the TSSs read counts was computed for each position in a window from −200 to +200 nucleotides for samples treated (blue curve) or not (red curve) with tobacco acid pyrophosphatase (TAP). The insert within the figure shows a zoomed view of the −100 to +50 nucleotide region. (**C**) Correlation between two biologically independent replicates (replicate 1: library L5p_03.WT, replicate 2: library L5p_04.WT; see *Table 1*). The Pearson's correlation value (ρ) between the read counts of the 174,151 TSSs identified in the two data sets is indicated.

The following figure supplements are available for figure 1:

**Figure supplement 1.** Reproducibility of transcription start site (TSS) sequencing.

**Figure supplement 2.** Transcription start site (TSS) consensus sequences.

(*Figure 1C*; Pearson's correlation coefficient (ρ) = 0.89, see also *Table 1*). When we compared data sets for the wild-type strain generated months apart in our laboratory (reproducibility test), we observed lower correlation coefficients (ρ = 0.71) likely reflecting biological variation inherent to samples prepared at different moments (*Figure 1—figure supplement 1A*). However, the correlation between data generated within our lab is still higher than the correlation between our data and results previously published by *Pelechano et al. (2013)* (ρ = 0.58; *Figure 1—figure supplement 1B*) or Arribere and Gilbert (ρ = 0.48; *Arribere and Gilbert, 2013*) as well as between these two sets of data (ρ = 0.55).

## TSS consensus sequences

The complementary DNA (cDNA) sequences were aligned with the genomic sequence (allowing one mismatch in the seed sequence; see 'Materials and methods'). Analysis of the TSSs and their surrounding sequences using the Web-LOGO algorithm (*Crooks et al., 2004*) identified a consensus

**Table 1**. Libraries generated and analysed in this study

| Library | Genotype | Total reads count | Unique reads count | Unique reads count mapped on *S. cerevisiae* genome | Unique reads count mapped on *S. pombe* genome |
|---|---|---|---|---|---|
| TSS-sequencing | | | | | |
| L5p_01 | WT | 8,650,655 | 5,862,245 | 5,631,307 | – |
| L5p_01 | upf1Δ | 9,624,626 | 7,161,842 | 6,733,121 | – |
| L5p_01 | rrp6Δ | 10,949,693 | 7,640,701 | 7,155,101 | – |
| L5p_01 | upf1Δrrp6Δ | 10,839,311 | 7,681,910 | 7,231,052 | – |
| L5p_02 | WT | 10,139,398 | 4,207,854 | 3,949,042 | – |
| L5p_02 | upf1Δ | 10,607,408 | 4,325,398 | 4,067,154 | – |
| L5p_02 | rrp6Δ | 22,861,396 | 10,269,822 | 9,711,133 | – |
| L5p_02 | upf1Δrrp6Δ | 13,803,020 | 6,526,805 | 6,207,357 | – |
| L5p_03 | WT | 11,525,631 | 4,670,586 | 4,343,264 | – |
| L5p_03 | upf1Δ | 9,832,590 | 3,338,679 | 3,122,942 | – |
| L5p_03 | rrp6Δ | 11,811,879 | 5,228,409 | 4,905,881 | – |
| L5p_03 | upf1Δrrp6Δ | 17,157,244 | 7,897,030 | 7,430,231 | – |
| L5p_04 | WT | 13,282,069 | 10,019,742 | 6,032,695 | – |
| L5p_04 | upf1Δ | 14,162,741 | 10,872,408 | 6,724,910 | – |
| L5p_04 | set2Δ | 14,714,187 | 11,048,398 | 6,591,345 | – |
| L5p_04 | upf1Δset2Δ | 16,958,167 | 12,121,894 | 8,081,113 | – |
| L5p_05 | WT | 11,270,824 | 4,446,988 | 4,172,618 | – |
| L5p_05 | upf1Δ | 12,093,631 | 4,599,307 | 4,323,962 | – |
| L5p_05 | set2Δ | 18,047,134 | 7,132,585 | 6,724,933 | – |
| L5p_05 | upf1Δset2Δ | 12,719,564 | 5,637,310 | 5,333,480 | – |
| L5p_06 | WT | 12,253,253 | 4,800,799 | 4,463,020 | – |
| L5p_06 | upf1Δ | 10,481,413 | 3,402,317 | 3,181,752 | – |
| L5p_06 | set2Δ | 12,179,190 | 4,448,762 | 4,167,876 | – |
| L5p_06 | upf1Δset2Δ | 14,269,228 | 5,584,019 | 5,227,982 | – |
| L5p_07 | upf1Δrrp6Δ + TAP | 8,890,286 | 2,973,612 | 1,621,712 | 1,134,673 |
| L5p_07 | upf1Δrrp6Δ − TAP | 13,836,172 | 3,214,470 | 149,092 | 2,700,018 |
| L5p_08 | upf1Δrrp6Δ + TAP | 9,689,188 | 2,768,418 | 1,341,975 | 1,209,108 |
| L5p_08 | upf1Δrrp6Δ − TAP | 9,765,672 | 2,390,794 | 82,685 | 2,074,096 |
| L5p_09 | upf1Δset2Δ + TAP | 11,885,938 | 3,793,668 | 1,976,161 | 1,555,818 |
| L5p_09 | upf1Δset2Δ − TAP | 11,105,585 | 2,936,749 | 117,202 | 2,552,264 |
| L5p_10 | upf1Δset2Δ + TAP | 11,665,986 | 3,797,708 | 1,343,091 | 2,147,100 |
| L5p_10 | upf1Δset2Δ − TAP | 11,000,429 | 2,476,135 | 59,450 | 2,145,439 |
| RNAseq | | | | | |
| LT_01 | WT | 8,104,047 | 7,257,423 | 6,634,522 | 1,761,046 |
| LT_01 | upf1Δ | 11,137,269 | 10,129,315 | 9,257,158 | 2,440,195 |
| LT_01 | xrn1Δ | 11,619,211 | 10,631,126 | 9,737,924 | 2,310,274 |
| LT_01 | upf1Δxrn1Δ | 7,947,627 | 7,299,151 | 6,645,353 | 1,678,281 |
| LT_02 | WT | 34,611,003 | 27,825,118 | 22,658,503 | 3,635,624 |
| LT_02 | upf1Δ | 29,379,233 | 24,421,717 | 19,815,913 | 2,966,963 |
| LT_02 | xrn1Δ | 26,816,267 | 22,686,420 | 18,245,251 | 2,493,471 |
| LT_02 | upf1Δxrn1Δ | 25,466,016 | 21,800,532 | 16,996,728 | 2,436,568 |

sequence around TSSs (*Figure 1—figure supplement 2A*) similar to the previously reported one derived from a smaller data set (*Zhang and Dietrich, 2005*). In particular, we observed a very strong bias to start at a purine (88% of mapped TSSs), usually following a pyrimidine (76% of the mapped TSSs), and the enrichment for an A at position −8 relative to the TSS (A(N)$_6$PyPu consensus). Surprisingly, 58% of TSS reads starting with a pyrimidine when aligned on the genome (12% of the mapped TSSs) show a mismatch at their first nucleotide, most of the time an A instead of the encoded pyrimidine (*Figure 1—figure supplement 2B*). Moreover, in 32% of these cases the surrounding genomic sequences exhibited a specific consensus, A(N)$_6$PyAAA (where the underlined base is the mapped TSS; *Figure 1—figure supplement 2C*). These observations suggest that, in these cases, transcription actually initiates on the A following the pyrimidine and that an additional A is added at the 5′-end of the transcript, possibly by a back-tracking mechanism as proposed in the model described in *Figure 1—figure supplement 2D*. This may also apply to other TSSs mapped on a pyrimidine and showing a mismatched first nucleotide. It thus appears from this observation that, even though 12% of the TSSs mapped on a pyrimidine, transcription initiation actually occurred on a pyrimidine in less than 5% of cases (42% of non-mismatched cDNAs out of 12% of cDNAs aligned on a pyrimidine).

## The nuclear and cytoplasmic RNA degradation pathways cooperate in the removal of pervasive transcripts

To assess the relative contributions of the cytoplasmic NMD and nuclear RNA control pathways in shaping the yeast transcriptome, the genes encoding Upf1, an RNA helicase essential for NMD, or Rrp6, a nuclear exosome catalytic subunit, were deleted (*He et al., 2003*; *Wyers et al., 2005*). Since, for a given gene, several closely spaced TSSs can be identified (*Pelechano et al., 2013*), we used a peak-calling procedure to define, in three biological replicates, TSS clusters corresponding to transcript isoforms with closely spaced 5′-ends (TSSCs; see 'Materials and methods'). This analysis allowed us to identify 17,812 TSSCs, among which 5927 could be assigned to 5′-ends of mRNAs corresponding to 5231 ORFs (O category in *Supplementary file 1*), as previously defined (*Pelechano et al., 2013*). Among the remaining TSSCs, 502 and 3644 were assigned to stable ncRNAs or repeated sequences (F category in *Supplementary file 1*) and previously described pervasive transcripts respectively (C, X and S categories for CUTs, XUTs and SUTs in *Supplementary file 1*; *Neil et al., 2009*; *Xu et al., 2009*; *van Dijk et al., 2011*).

In contrast to mRNA-TSSCs, the majority of which were unaffected in the absence of Upf1, Rrp6 or both (*Figure 2A*) and, as expected (*Wyers et al., 2005*), CUT-TSSCs were strongly stabilized in the absence of Rrp6 (*Figure 2B*; 79% stabilized significantly, as determined using the moderated estimation of fold change and dispersion function of DESeq2—*Love et al., 2014*; see 'Materials and methods'). In contrast, only 23% of the CUT-TSSCs increased significantly in the absence of Upf1. The opposite situation was observed for XUTs and SUTs, which were more sensitive to the cytoplasmic NMD quality-control pathway (28% vs 52% significantly stabilized in the absence of Rrp6 or Upf1, respectively; *Figure 2C,D* and *Supplementary file 1*). Collectively, 39% and 54% of the pervasive transcripts (CUTs, SUTs and XUTs) were significantly sensitive to the absence of Upf1 in a wild-type or *rrp6Δ* background, respectively, indicating that NMD targets about half of the pervasive transcripts. The absence of both Rrp6 and Upf1 had an additive effect on the stabilization of pervasive transcripts. In the double mutant, 76% and 92% of the XUTs/SUTs and CUTs, respectively, were significantly stabilized, suggesting that NMD provides a fail-safe control mechanism for pervasive transcripts that escaped degradation in the nucleus, reminiscent of what has been previously shown for some unspliced pre-mRNAs (*Sayani and Chanfreau, 2012*).

## XUTs and SUTs carry short spurious ORFs that target them for degradation by NMD

The accumulation of XUTs in the absence of Upf1 suggests that these transcripts, originally described as Xrn1-sensitive (*van Dijk et al., 2011*), may be primarily targeted for degradation by NMD. We tested this hypothesis by performing northern blot hybridization for three XUTs (*Figure 3A*) and genome-wide transcriptome analyses of wild-type, *upf1Δ*, *xrn1Δ*, and *upf1Δxrn1Δ* cells (*Figure 3B–D*). Since the absence of Xrn1 leads to the accumulation of decapped RNAs (*Hsu and Stevens, 1993*), we could not use the TSS sequencing methodology applied for the other mutants and we thus used, for this particular experiment, a classical RNA sequencing approach for transcript quantification

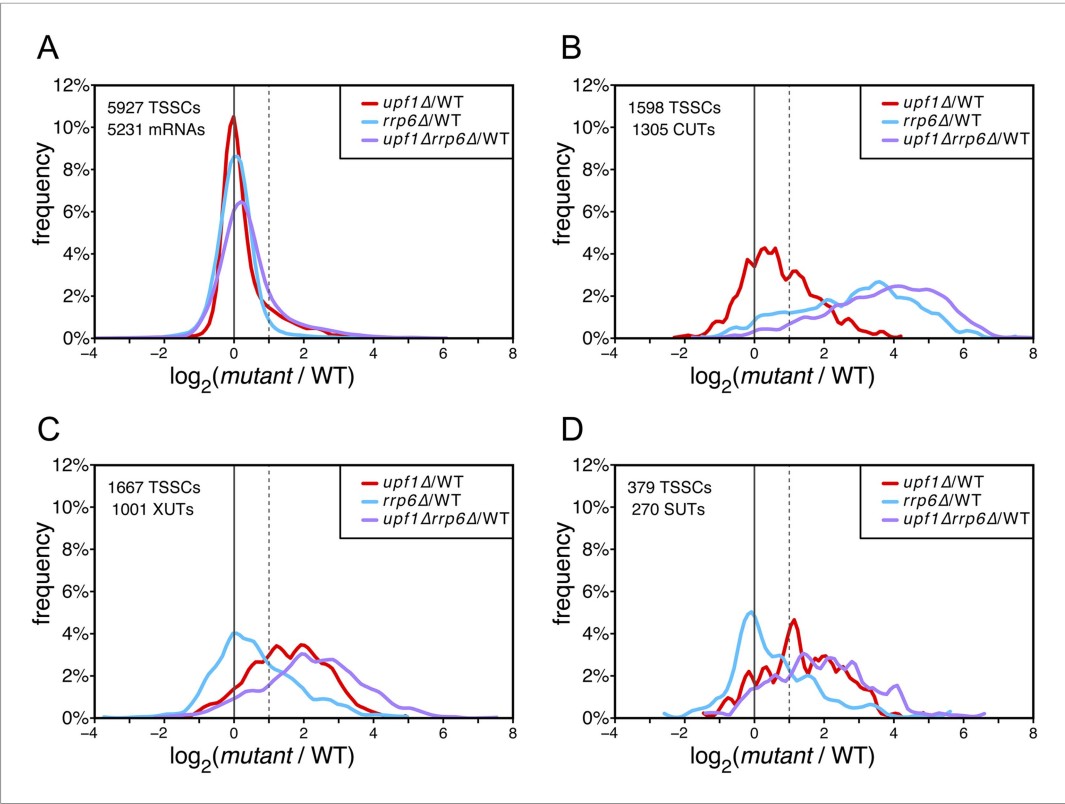

**Figure 2**. Differential effect of *UPF1* and/or *RRP6* deletion on mRNAs and pervasive transcripts. Frequency distribution of the ratios of transcription start site clusters (TSSCs) read counts in *upf1Δ* (red), *rrp6Δ* (blue), or *upf1Δrrp6Δ* (purple) compared with wild-type (WT) for mRNAs (**A**), cryptic unstable transcripts (CUTs) (**B**), Xrn1-sensitive transcripts (XUTs) (**C**) and stable unannotated transcripts (SUTs) (**D**). The dashed vertical lines mark a twofold increase in TSSC read counts in the various mutants relative to the wild-type. The number of identified TSSCs and of features to which they were assigned is indicated for each transcript class. Note that CUTs, XUTs and SUTs constituting overlapping transcript populations, when a TSSC was assigned to a pervasive transcript annotated in more than one of these classes, we arbitrarily associated the corresponding TSSC in priority to CUTs, then to XUTs and finally to SUTs.

(see 'Materials and methods'). Moreover, since the absence of Xrn1 affects the overall cellular mRNA content (*Sun et al., 2013*), an aliquot of a *S. pombe* culture was added to the cell pellet before RNA extraction to provide an independent internal control for normalization of the results. After having verified that the overall quantifications with 'RNAseq' gave results similar to those obtained by TSS sequencing for the wild-type and *upf1Δ* strains (*Figure 3B* and *Supplementary file 2*), we used the former technique to analyse the genome-wide effect of the *XRN1* deletion. Consistent with previous reports (*van Dijk et al., 2011*; *Sun et al., 2013*), deletion of *XRN1* resulted in a global increase of mRNAs, XUTs and SUTs compared with the wild-type (*Figure 3C,D*). In contrast to mRNAs, the extent to which XUTs and SUTs were stabilized in *xrn1Δ* was very similar to the one observed in *upf1Δ* cells (*Figure 3C,D*). Furthermore, deleting *XRN1* in an *upf1Δ* background had almost no additional stabilizing effect on XUTs and SUTs, in contrast to mRNAs. This epistatic relationship observed between *xrn1Δ* and *upf1Δ* for the stabilization of XUTs and SUTs is consistent with these transcripts being primarily targeted by NMD, and with Xrn1 acting as a downstream effector of this pathway.

In contradiction to the fact that XUTs and SUTs have been designated as 'non-coding', the strong effect NMD inactivation had on these transcripts indicates that they must, at some point, be translated. Indeed, all XUTs and SUTs carry spurious ORFs (*Figure 3—figure supplement 1A*) and some of them have been identified in ribosomal profiling experiments and shown to encode short peptides (*Ingolia et al., 2009*; *Smith et al., 2014*). However, the high sensitivity to NMD of XUTs and SUTs compared with mRNAs suggests the presence of specific features. Assuming that the first

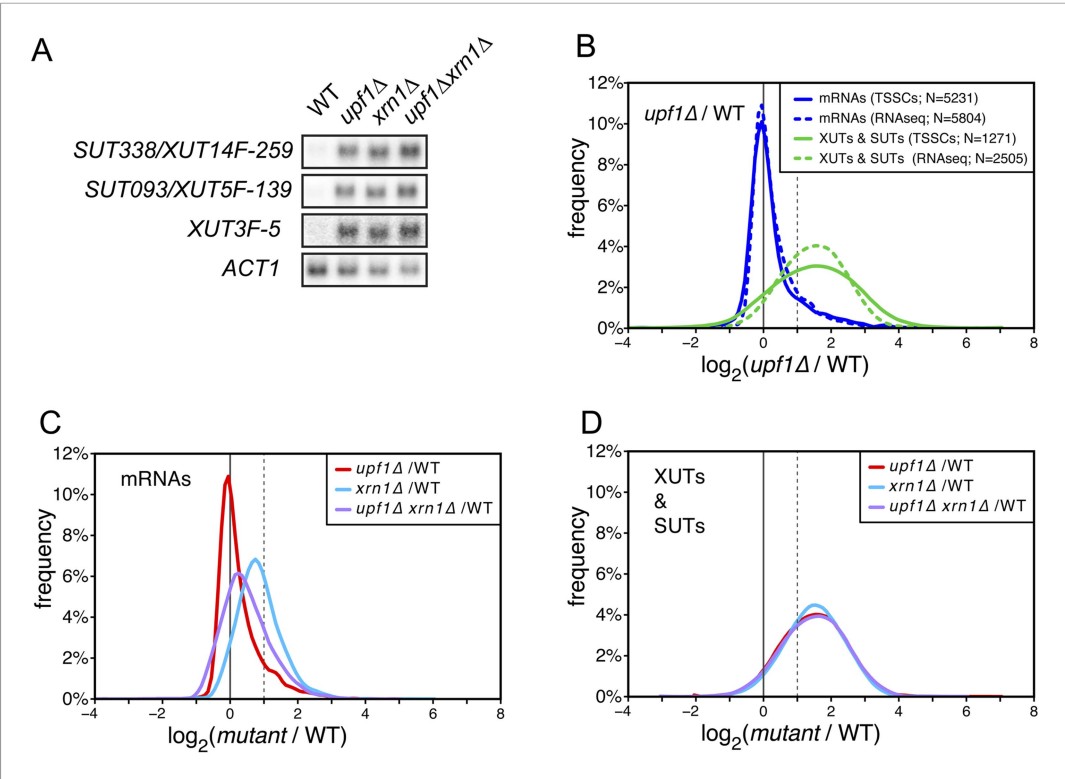

**Figure 3**. XUTs are primarily targeted for degradation by the nonsense-mediated mRNA decay (NMD). (**A**) Northern blot analysis of three different Xrn1-sensitive transcripts (XUTs) in wild-type (WT), *upf1Δ*, *xrn1Δ*, and *upf1Δxrn1Δ* cells. *ACT1* was used as a loading control. (**B**) Frequency distribution of the ratios of transcription start site clusters (TSSCs) read counts between *upf1Δ* and wild-type cells obtained with two different methods for library preparation and normalization procedures. 'TSSCs' refers to data obtained using the protocol developed to identify TSSs and 'RNAseq' to the protocol used to obtain the whole transcriptome (see 'Materials and methods'). Only reads corresponding to annotated mRNAs, XUTs and SUTs were included in the analysis (see *Supplementary file 2*). (**C**) and (**D**) Frequency distribution of the ratios of read counts for *upf1Δ* (red), *xrn1Δ* (blue) or *upf1Δxrn1Δ* (purple) compared with wild-type for mRNAs and XUTs and SUTs respectively. The vertical dashed lines mark a twofold increase in read counts.

The following figure supplement is available for figure 3:

**Figure supplement 1**. The presence of short open reading frames (ORFs) and long 3′-UTRs is a hallmark of natural nonsense-mediated mRNA decay (NMD) substrates.

---

encountered ORF is translated, XUTs and SUTs carry, on average, much shorter ORFs and longer 3′-UTRs than mRNAs (*Figure 3—figure supplement 1A*). The presence of both a long 3′-UTR and a short ORF increases NMD efficiency in budding yeast (*Decourty et al., 2014*) and thus explain why these transcripts are efficiently targeted for degradation by this pathway. Furthermore, the few XUTs and SUTs found not to be sensitive to NMD have on average significantly shorter 3′-UTRs (*Figure 3—figure supplement 1B*), which is in agreement with the recently published observation that, in ribosomal profiling experiments, the length of RNA downstream of the ribosome protected region is significantly longer for NMD-sensitive compared with NMD-insensitive unannotated RNA transcripts (*Smith et al., 2014*).

## Identification of additional pervasive transcripts

Amongst the 17,812 TSSCs identified in this study, 7739 could not be assigned to previously annotated transcripts (ORFs, stable ncRNAs or pervasive transcripts) and were only detected in mutants cells (*Figure 4*). These newly identified transcripts, expressed at low levels (*Figure 4—figure*

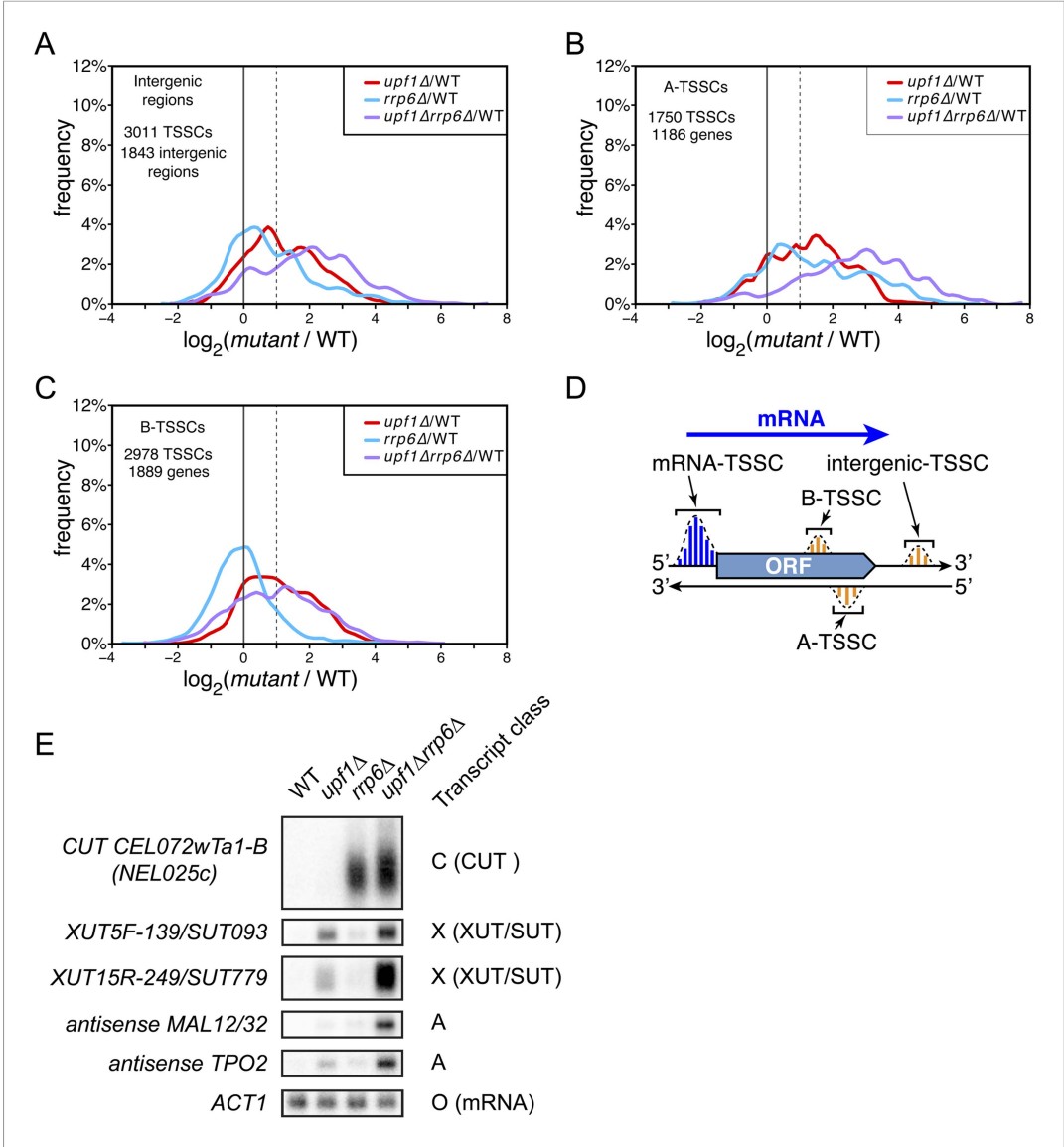

**Figure 4.** Novel transcripts revealed upon deletion of UPF1 and/or RRP6. (A–C) Frequency distribution of the ratios of transcription start site clusters (TSSCs) read counts in *upf1Δ* (red), *rrp6Δ* (blue) or *upf1Δrrp6Δ* (purple) compared with wild-type for transcripts initiating within intergenic regions—'intergenic TSSCs' (A), from within an mRNA transcribed region but antisense to the mRNA—A-TSSCs (B) or within an mRNA transcribed region, but in the sense orientation with respect to the mRNA—B-TSSCs (C). (D) Schematic representation of the various classes of TSSCs described above. The small blue and orange vertical bars represent individual TSSs within TSSCs (dashed lines). (E) Northern blot analysis of poly(A)+ RNA from wild-type, *upf1Δ*, *rrp6Δ* and *upf1Δrrp6Δ*. The category to which the transcripts belong is indicated on the right. *ACT1* was used as a loading control.

The following figure supplements are available for figure 4:

**Figure supplement 1.** Distribution of read counts for different classes of transcripts.

**Figure supplement 2.** Transcription start site (TSS) consensus sequences for the different classes of transcripts.

---

*supplement 1*), originated from intergenic regions (3011 'intergenic-TSSCs' originating from 1843 intergenic regions) as well as from within mRNA transcribed regions either in sense (2978 'B-TSSCs' within 1889 mRNAs) or in antisense orientation (1750 'A-TSSCs' antisense to 1186 mRNAs; *Supplementary file 1*). While 'B-TSSCs' were sensitive to *upf1Δ* and almost not affected by the

deletion of *RRP6* (*Figure 4C* and *Figure 4—figure supplement 1*), 'intergenic' and 'A-TSSCs' were affected by the two mutations (*Figure 4A,B* and *Figure 4—figure supplement 1*). Furthermore, the absence of both Rrp6 and Upf1 had an additive effect on the accumulation of these last two classes, as is the case for previously identified pervasive transcripts. Combining these two mutations might sometimes even have a synergistic and not only an additive effect, since some of these transcripts were readily detectable only in the *upf1Δrrp6Δ* double mutant (e.g., the transcript found antisense to *MAL12/32* in *Figure 4E*). Analysis of individual TSSs signatures for these three classes of previously unannotated transcripts using the Web-LOGO algorithm (*Crooks et al., 2004*) yielded a consensus sequence almost indistinguishable from the one obtained for individual TSSs assigned to known mRNAs or previously identified pervasive transcripts (*Figure 4—figure supplement 2*), suggesting that the underlying DNA sequence plays an important role in TSS selection by the scanning polymerase.

## Co-transcriptional histone modifications and NMD cooperate to restrict the accumulation of internally initiated transcripts

Even though transcripts initiated inside ORFs are expected to be targeted for degradation by NMD due to the presence of short spurious ORFs, the high number of internally initiated transcripts ('A and B-TSSCs') identified in cells lacking *UPF1* was surprising. The synthesis of such transcripts is normally repressed within transcribed regions unless the proper chromatin structure cannot be re-established in the wake of RNAPII (*Smolle and Workman, 2013*). In particular, histone methylation by Set2 was shown to be a key determinant of this repression. We therefore analysed the impact of *SET2* deletion on 'A' and 'B' TSSCs identified in the *upf1Δ* strain. While deletion of *SET2* had little effect on the internal TSSCs identified in the single *upf1Δ* mutant, it revealed new Set2-sensitive 'A' and 'B' TSSCs (*Figure 5* and *Supplementary file 3*). As expected and in contrast to A-TSSCs and B-TSSCs, deletion of *SET2* had no global effect on ORF TSSCs and only a marginal effect on CUTs, XUTs and SUTs or 'intergenic' TSSCs (*Figure 5—figure supplement 1*). In agreement with the observed enrichment in Set2-catalyzed H3K36 methylation towards the 3′ end of ORFs (*Pokholok et al., 2005*), the 'A' and 'B' TSSCs more sensitive to Set2 were located further away from the mRNA 5′-ends (*Figure 6A–C*). Analysis of the sequence surrounding individual TSSs from these TSSCs identified a pattern almost identical to the one found for ORF-TSSs (*Figure 6—figure supplement 1*), confirming that they corresponded to *bona fide* transcription initiation events. However, unlike the TSSs associated with other features identified in this study, the 'B' TSSs were not associated with a strong NFR

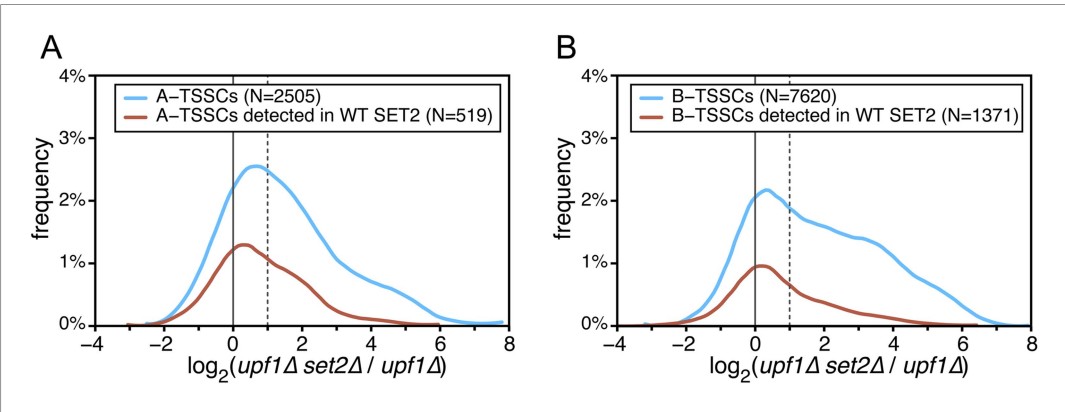

**Figure 5**. Effect of the absence of *SET2* on transcription start sites (TSSs) identified inside open reading frames (ORFs). (**A**) Frequency distribution of the ratios of TSS clusters (TSSCs) read counts in *upf1Δset2Δ* compared with *upf1Δ* cells for the intragenic A-TSSCs. The comparison was performed for all the TSSCs (blue line) and for the ones identified in a single *upf1Δ* mutant (red line). (**B**) As in (**A**) but for B-TSSCs.

The following figure supplement is available for figure 5:

**Figure supplement 1**. Deletion of *SET2* specifically increases the expression level of intragenic transcription start site clusters (TSSCs).

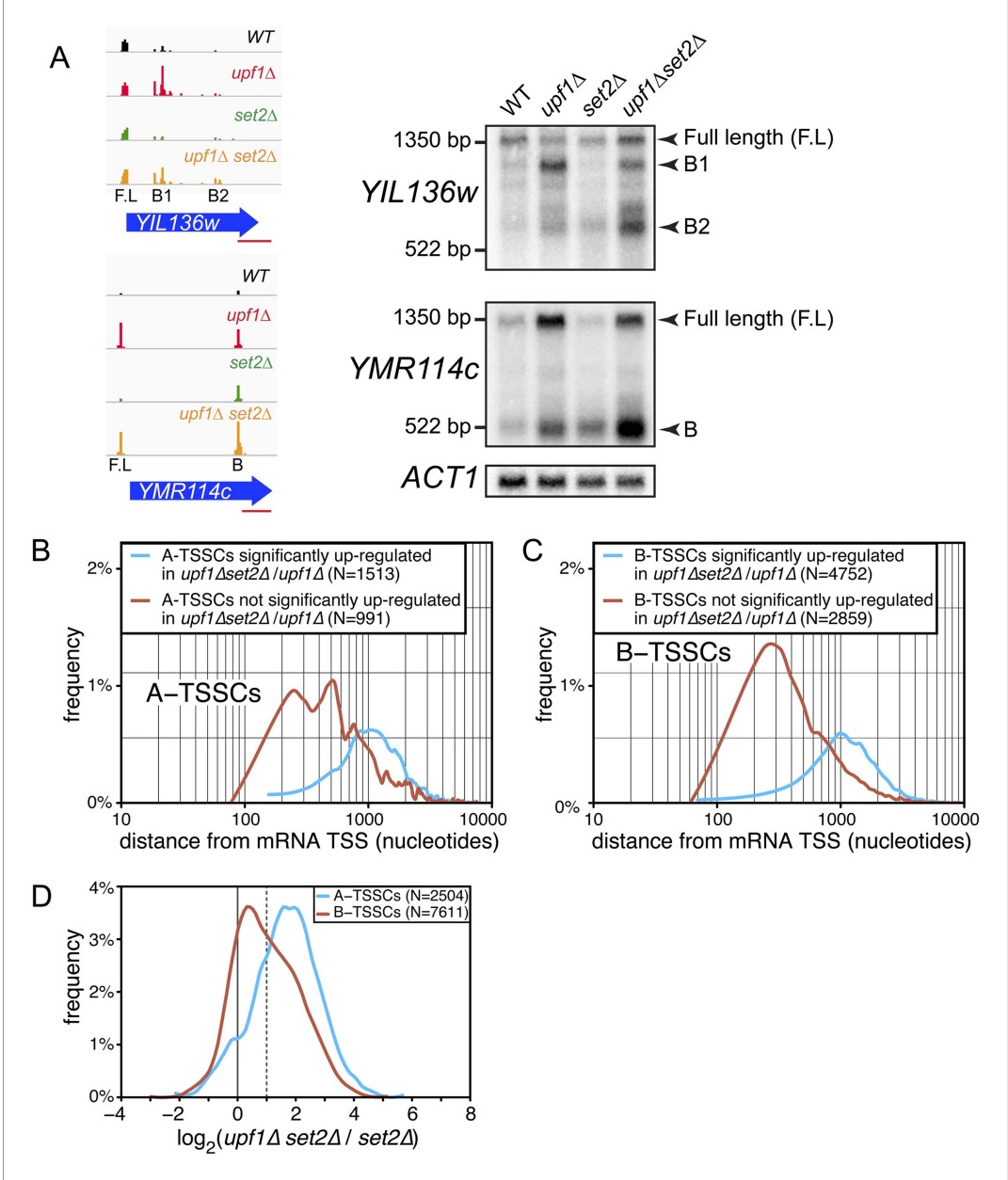

**Figure 6**. Transcription start site (TSSs) identified inside open reading frames (ORFs) show a differential sensitivity to the absence of *SET2* according to their position along the mRNA. (**A**) Visualization of the TSS reads at the *YIL136w* and *YMR114c* loci in the wild-type (WT; black), *upf1Δ* (red), *set2Δ* (green), and *upf1Δset2Δ* (orange) cells. The blue arrows represent ORFs and the horizontal red bar the position of the probes used for the Northern blots displayed on the right. Arrowheads in the right panel indicate the position of the full-length and internally initiated (B1, B2, and B) transcripts. (**B**) and (**C**) Frequency distribution of A-TSSCs and B-TSSCs read counts respectively in *upf1Δset2Δ vs upf1Δ* according to the distance from their associated mRNA TSS. Blue and red lines are for TSSCs sensitive and insensitive to the deletion of *SET2* respectively. (**D**) Frequency distribution of read counts for the A- (blue) and B- (red) TSSCs in *upf1Δset2Δ* compared to *set2Δ* cells.

The following figure supplements are available for figure 6:

**Figure supplement 1**. Consensus sequences around the transcription start sites (TSSs) for (**A**) A-TSSCs and (**B**) B-TSSCs identified in the *upf1Δset2Δ* mutant generated using the Web-LOGO algorithm (*Crooks et al., 2004*).

**Figure supplement 2**. Transcription start sites (TSSs) are associated within nucleosome-free regions.

(*Figure 6—figure supplement 2*). Instead, analysing the nucleosome density around these TSSs revealed the presence of a weak NFR surrounded by two well-positioned nucleosomes. This particular pattern might explain the high sensitivity of 'B' TSSs to mutations affecting chromatin structure, such as the deletion of the Set2 histone methyl-transferase (*Smolle and Workman, 2013*).

Importantly, 55% of the A-TSSCs and 40% of the B-TSSCs, whether repressed by Set2 or not, were sensitive to Upf1 (i.e., increased significantly in *upf1Δset2Δ* compared with *set2Δ* cells; *Figure 6D* and *Supplementary file 3*), revealing NMD to be an important quality-control mechanism to eliminate internally initiated transcripts.

## Almost half of the coding genes produced transcript isoforms sensitive to NMD

In contrast to their strong impact on the steady state levels of a variety of pervasive transcripts, including previously unannotated ones, inactivation of nuclear and/or cytoplasmic RNA quality-control pathways had a relatively minor global effect on the steady state levels of coding transcripts (*Figure 2*).

Consistent with previously reported data (*He et al., 2003*; *Wyers et al., 2005*), the effect on mRNAs was larger upon deletion of *UPF1* (with 17% of the mRNAs-TSSCs showing a significant increase in the absence of Upf1) than upon deletion of *RRP6* and deletion of both genes simultaneously did not show any additive effect (*Figure 2A*). Yet, due to the mRNA 5′-end heterogeneity, analysing individual TSSs gave a different picture. Indeed, transcript isoforms in which TSSs where followed by uORFs were, globally, stabilized in the absence of *UPF1*, while those in which the annotated start codon was the first ATG downstream the TSS were mostly unaffected (*Figure 7A*, solid lines; *Figure 7B*). A large fraction of these NMD-sensitive transcripts corresponded to minor isoforms, explaining why NMD had only a weak effect on the overall mRNA-TSSC levels (*Figure 2A*). Yet, for 1129 out of the 5231 active genes (22%), a fraction of their transcript isoforms carried at least one uORF and was significantly sensitive to NMD. Isoforms carrying more than one uORF appeared even more sensitive to NMD (*Figure 7A*), suggesting that the first AUGs of uORFs containing transcripts are not always efficiently used for translation initiation, likely because they are not in a favorable context (*Arribere and Gilbert, 2013*).

Unexpectedly, another important category of transcript isoforms found to be sensitive to NMD corresponded to TSSs mapping downstream to the annotated ORF ATG (here called iTSSs; *Supplementary file 4*). Only observed after TAP treatment (see insert in *Figure 1B*), these iTSSs must correspond to genuine capped-RNAs. We considered genes as having iTSSs (3327 genes; 64% of the expressed genes) if they contained at least four iTSS reads, which accounted for 99% of all iTTS reads. A fraction of transcript isoforms initiated at iTSSs followed by an out-of-frame ATG and significantly stabilized in the absence of Upf1 could be identified for 1821 out of the 5231 expressed genes (35%, *Figure 7A,B*). In contrast to the weak overall effects observed for TSSs located directly upstream of the start codon, the mean expression level of iTSSs significantly increased in the *upf1Δ* strain relative to the wild-type (*Figure 7C*, and *Figure 7—figure supplement 1*). Note that RNAPII transcription initiating preferentially at purines, the A and G of the annotated start codons are quite frequently used as sites of transcription initiation and the first nucleotide of codons are preferential sites of internal initiation because these positions are enriched in purines (*Figure 7—figure supplement 1B,C*; *Mackiewicz et al., 1999*). The consensus sequence for the iTSSs was not different from the general consensus sequence for RNAPII transcription initiation (*Figure 7—figure supplement 2*).

Altogether, transcript isoforms significantly stabilized in NMD-deficient cells, either because they carry a uORF or because they initiated at an out-of-frame iTSS, were found in 2437 genes; that is, 47% of the 5231 expressed genes.

In transcripts initiated at iTSSs, the first encountered ATGs are most of the time out-of-frame relative to the main ORF (*Figure 8A*). The first ORFs of the corresponding transcripts are thus short and followed by a long 3′-UTR, making them excellent NMD substrates. This bias towards out-of-frame ATGs was specifically observed at the beginning of genes for which iTSSs were identified (*Figure 8B,C*). This correlated with a significantly lower frequency of methionine in the N-terminal part of the corresponding proteins (*Figure 8D*), even though methionines were found to be globally underrepresented at the beginning of all yeast proteins. The frequent use of iTSSs by RNAPII thus does not generally result in the production of truncated proteins and tends to generate transcripts that are sensitive to NMD.

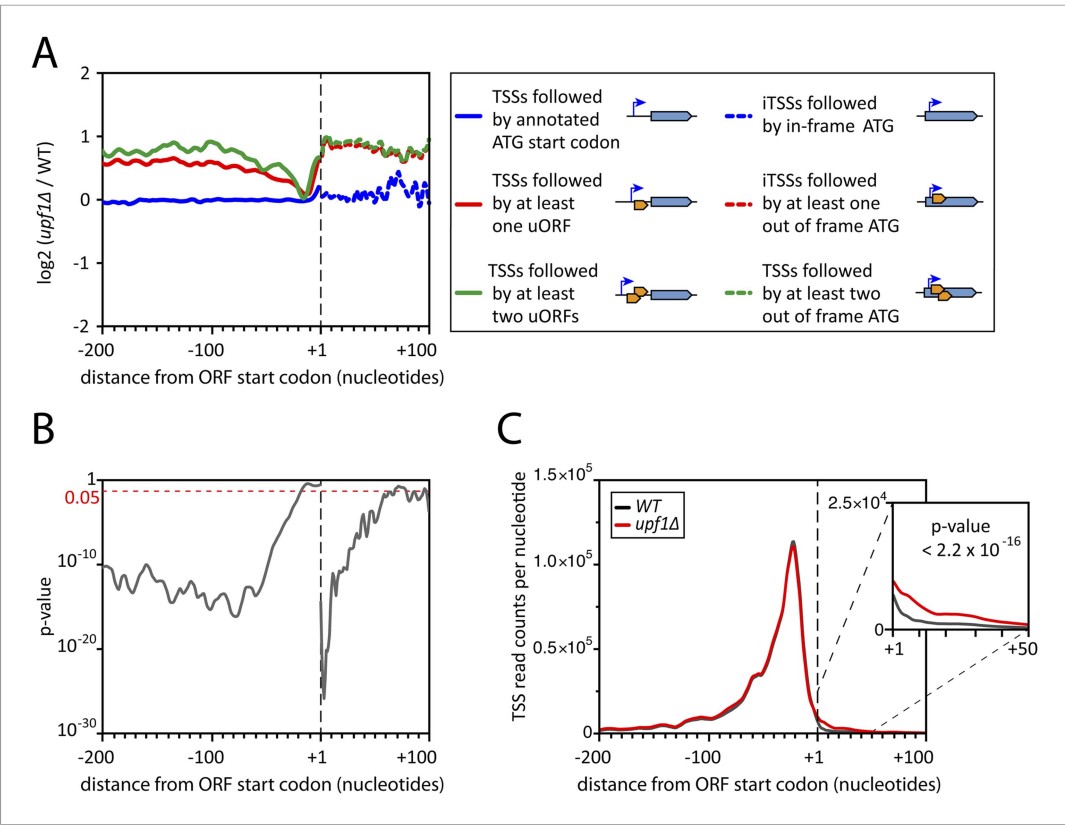

**Figure 7**. Deletion of *UPF1* reveals numerous minor mRNA-associated transcription start sites (TSSs). (**A**) Genes were aligned by their start codon and the $\log_2$ of ratios of TSS reads for *upf1Δ* vs wild-type was plotted for TSSs upstream or downstream (iTSSs) the annotated ATG start codons, as depicted in the right panel. The main open reading frames (ORFs) are represented by large blue arrows and upstream ORFs (uORFs) or small internal out-of-frame ORFs by small orange arrows. The thin blue arrows indicate the TSSs. (**B**) The curve represents the probability at each nucleotide position that the distributions of reads corresponding to the red and blue curves shown in **A** are the same. The dashed red line marks the 0.05 p-value. (**C**) Genes were aligned by their annotated start codons (A of the ATG at position +1) and the cumulative TSS read counts per nucleotide (smoothed over 11 nucleotides) was plotted for the wild-type (black) and *upf1Δ* (red) cells. Inset: Magnification of the +1 to +50 region. The p-value $< 2.2 \times 10^{-16}$ is the probability (ANOVA test) that the distributions of the values, per nucleotide, for the red and black curves are the same within the +1 to +50 nucleotides interval.

The following figure supplements are available for figure 7:

**Figure supplement 1**. Initiation of iTSS at purines reflects a bias in codon composition.

**Figure supplement 2**. (iTSSs) consensus sequences.

## Discussion

### Genome-wide identification of TSSs

We describe here the use of a modified 5′-RACE technique to identify the 5′-ends of capped RNA, which map RNAPII TSSs. The method, which includes a streptavidin-biotin purification step, is highly specific and can be applied to any eukaryotic organism. Our results provide the most comprehensive genome-wide identification of TSSs in budding yeast, not only for mRNAs but also for a wide range of pervasive transcripts, including previously non-annotated ones. This was made possible by combining this highly specific TSS determination technique with the use of mutants affecting both the nuclear exosome and the cytoplasmic NMD pathway.

Detailed analyses of the sequencing reads and the corresponding upstream genomic sequences extended and validated the previously identified sequence consensus surrounding TSSs in *S.*

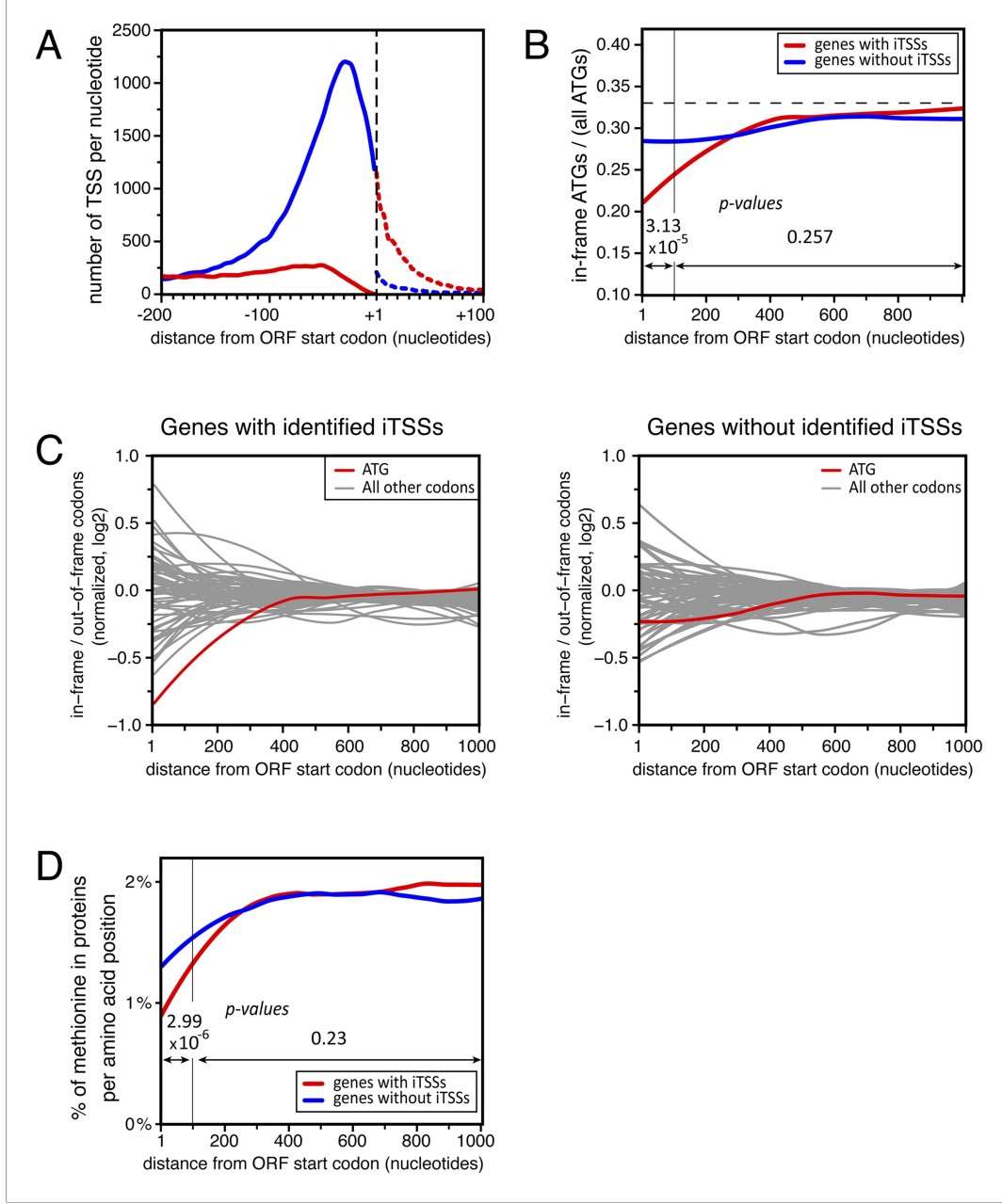

**Figure 8**. Distribution of in-frame vs out-of-frame ATG codons and methionines for genes with or without transcription start sites downstream of annotated ATGs (iTSSs). (**A**) Cumulative total number of transcription start sites (TSSs) per nucleotide in *upf1Δ* cells for TSSs directly followed by the annotated ATGs (blue line) or a upstream open reading frame (uORF) (red line) and for TSSs downstream the annotated ATGs (iTSSs) and followed by an out-of-frame ATG (red dashed line) or by an in-frame ATG (blue dashed line). (**B**) Proportion of in-phase ATGs (+1 frame) following an annotated start codon, binned over nine nucleotides, for protein-coding genes with (red; 3327 genes) or without (blue; 1904 genes) iTSS reads in *upf1Δ*, as defined in the text. The dashed line indicates the expected value for a random distribution. Genes with iTSSs reads are significantly depleted of in-frame ATGs relative to all ATGs in the first 100 nucleotides when compared with genes without iTSSs reads (p-value = 3.13 10$^{-5}$; ANOVA). (**C**) Proportion (log$_2$) along the ORFs of in-frame codons, normalized over the regions downstream the first 300 nucleotides of genes, for all codons (binned over nine nucleotides) for the two sets of genes defined in **B**. The ATG codon is in red. (**D**) Frequency of methionines per amino acid position along the ORFs for the two sets of genes defined in (**B**). The p-values (ANOVA method) for the difference in methionine composition over the regions between 0–100 and 100–1000 nucleotides are indicated on the figure.

*cerevisiae* (A(N)$_6$PyPu; *Figure 1—figure supplement 2A* and *Figure 4—figure supplement 2*, and *Zhang and Dietrich 2005*). Importantly, this consensus sequence was observed for the TSSs of all the different classes of transcripts identified in this study. Note that 1,775,474 sites in the genome conform to this consensus but only 85,714 (4.8%) were used as TSSs.

We also identified cases in which transcription started on what appears to be a 'slippery' sequence (PyAAA), which resulted in the incorporation of an additional non-encoded A at the 5′-end of the transcripts. The simplest explanation for this observation is that the very short nascent RNA formed after two or three nucleotide incorporation is able to shift back by one nucleotide, probably together with the RNAPII, before transcription resumes (*Figure 1—figure supplement 2D*), suggesting that the ternary complex formed by the transcribing polymerase, the RNA moiety and chromatin is rather labile at this early stage of transcription.

## Role of nonsense-mediated RNA decay in removal of pervasive and cryptic transcripts

Our analyses of TSSs in NMD-deficient cells revealed the prevalent role of this cytoplasmic RNA quality-control pathway in eliminating pervasive transcripts. We found that 52% of the SUTs and XUTs were significantly stabilized in *upf1Δ* cells when using the TSSCs from the TSS-sequencing experiments (see *Supplementary file 1*). When using the RNAseq approach, this proportion was even higher (70%; see *Supplementary file 2*), possibly because of the overall higher sequence read counts for a given transcript, which provides more power to statistical analyses. Since NMD is a translation-coupled RNA degradation pathway, the stabilizing effect seen upon *UPF1* deletion on pervasive transcripts indicated that once these transcripts reached the cytoplasm they associate with the translation machinery. Indeed, once they reached the cytoplasm the pervasive transcripts (being capped and poly-adenylated) cannot be distinguished from normal mRNAs and associate with the translation machinery. However, since they usually have short ORFs followed by relatively long 3′ UTR regions (*Figure 3—figure supplement 1*), a hallmark of NMD substrates in yeast (*Decourty et al., 2014*), they will be targeted for degradation by this RNA quality-control pathway. This is supported by the identification of pervasive transcripts (mainly XUTs and SUTs) in ribosomal profiling experiments performed in wild-type cells and the recent discovery of a new class of unannotated transcripts (uRNAs) associated with polyribosomes and encoding short peptides (*Ingolia et al., 2009*; *Smith et al., 2014*).

The association of so-called 'non-coding' transcripts with the translation machinery and their ensuing high sensitivity to NMD has also been observed in multicellular organisms. In mouse embryonic stem cells, many long non-coding RNAs (lncRNAs) were found to be associated with translating ribosomes and an estimated 17.4% of lncRNAs were found to be upregulated in the absence of *UPF1*, compared with only 4% for protein-coding genes (*Ingolia et al., 2014*; *Smith et al., 2014*).

Our results also showed that XUTs, originally described as Xrn1-sensitive transcripts (*van Dijk et al., 2011*), are in fact primarily targeted by NMD by virtue of their poor coding capacities, with the Xrn1 exonuclease acting as a downstream effector of this pathway. Thus, we propose to redefine the various classes of pervasive transcripts identified in yeast into transcripts primarily degraded in the cytoplasm (SUTs and XUTs) and transcripts more sensitive to the nuclear exosome (CUTs) even though each of these degradation pathways similarly affect some transcripts and can show a synergic effect (see below).

Our analysis also revealed the prevalent role of NMD in the removal of cryptic transcripts initiating from intragenic promoters either in sense (here called B) or in antisense orientation (here called A). The strong impact of NMD inactivation on these transcripts indicated that they are exported to the cytoplasm and associate with the translation machinery, as is the case for the pervasive transcripts (see above). Even though we cannot rule out an indirect effect of the *UPF1* deletion on chromatin structure, the high number of intragenic transcripts (whether sense or antisense) identified in the *upf1Δ* single mutant suggests that the inhibition of transcription initiation by transcription-coupled chromatin modifications is not fully efficient and thus that some of these transcripts are produced as part of the normal transcription cycle in wild-type cells, with NMD playing an important role to eliminate them.

## Cooperation between nuclear and cytoplasmic RNA quality-control pathways

Deletions of *UPF1* and *RRP6* had an additive effect on the accumulation of CUTs, SUTs and XUTs. However, a number of pervasive transcripts, in particular 'A' and 'intergenic' ones, were stabilized to

substantial levels only when both the nuclear and the cytoplasmic RNA quality-control pathways were compromised, suggesting that they can act synergistically.

Two mechanisms of transcription termination have been reported in yeast. One depends on the cleavage and poly-adenylation complex (CPF-CFI/II), which generates the poly-adenylated mRNAs that get exported and translated in the cytoplasm, and another, which involves the NNS complex and is shared by CUTs and sn(o)RNAs precursors (see 'Introduction'). However, the demarcation between the two modes of termination is far from being strict as some terminators can often be recognized by both pathways depending on their distance from the TSS (*Porrua et al., 2012*). In the early phase of transcription elongation the RNAPII CTD repeats are mainly phosphorylated at Ser5, which favor recruitment of the NNS complex; while transcription proceeds, Ser2 gets phosphorylated at the expense of Ser5, promoting the recruitment of the CPF-CFI/II complex (*Ahn et al., 2004*; *Kim et al., 2004*). For some pervasive transcripts, these two termination pathways may compete and generate transcripts terminated by the NNS pathways and degraded by the nuclear exosome, as well as transcripts terminated by the CPF-CFI/II pathway and exported to the cytoplasm where they are targeted for degradation by the NMD. The produced RNAs would thus accumulate to detectable levels only when both RNA quality-control pathways are inactivated. Since the nuclear exosome has recently been shown to act with the NNS complex to promote early transcription termination at specific targets in *S. cerevisiae* (*Fox et al., 2015*), we cannot rule out that in wild-type or *upf1Δ* cells transcription will normally be terminated by the NNS complex and the synthesized transcripts degraded by the nuclear exosome, while in *rrp6Δ* cells transcription will proceed until the polymerase encounters the next CPF-CFI/II termination signal giving rise to longer transcripts exported to the cytoplasm and targeted for degradation by the NMD.

## NMD removes numerous transcripts arising from the low specificity of transcription initiation

NMD was previously shown to target a few hundred uORF-containing mRNAs (*He et al., 2003*; *Guan et al., 2006*; *Johansson et al., 2007*; *Arribere and Gilbert, 2013*). However, the use of NMD-deficient cells allowed us to identify 22% of the active genes for which a fraction of their transcript isoforms carried at least one uORF and were significantly sensitive to NMD. Moreover, 35% of the expressed genes generated transcript isoforms initiating at iTSSs and significantly stabilized in the absence of Upf1 (see *Figure 7A,B*). Some transcripts initiated at iTSSs were previously described in wild-type cells and shown to allow the synthesis of N-terminal variants of proteins exhibiting differential stabilities or localizations (*Wu and Tzagoloff, 1987*; *Gammie et al., 1999*; *Arribere and Gilbert, 2013*; *Pelechano et al., 2013*) but their number was vastly underestimated, probably because of their sensitivity to NMD (see *Figure 7C*). These two observations reveal an important role for NMD in getting rid of numerous undesired transcripts, the majority of which likely resulted from the low specificity of TSS selection by RNAPII. Yet, we cannot rule out that some of these transcripts have a biological function. For example, the use of alternative TSSs giving rise to transcripts with very different sensitivity to NMD could be used for regulatory purposes, in a way similar to that described for some genes of the nucleotide biosynthetic pathway (*Kuehner and Brow, 2008*; *Thiebaut et al., 2008*).

Altogether, transcript isoforms carrying a uORF or starting at an iTSS and targeted by NMD constituted the major transcript isoforms for only 446 mRNAs and minor transcript isoforms for almost half of the expressed genes. Therefore, although not quantitatively impacting the overall mRNA levels substantially (*Figures 2A, 7C*), NMD affects qualitatively a large fraction of the mRNA transcription units. NMD also has a strong impact on the accumulation of pervasive and cryptic transcripts and thus appears as a major player shaping the yeast transcriptome. In addition, we observed that transcripts initiated at iTSSs are significantly depleted in ATGs in the +1 frame, precluding the synthesis of N-terminally truncated proteins and ensuring their efficient NMD degradation. It thus suggests that this phenomenon might be important enough to have imposed an evolutionary constraint on the methionine content of the N-terminus of the yeast proteins.

## Materials and methods

### Yeast strains and culture

All the strains are a derivative of BY4741 and were obtained directly from the Euroscarf deletion collection (http://web.uni-frankfurt.de/fb15/mikro/euroscarf/) or generated by crossing with a *can1Δ*

derivative of BY4741 (LMA1057, see *Table 2*). Cells were grown to mid-exponential phase in YPD-rich medium at 30°C in a microturbidostat as previously described (*Decourty et al., 2008*), harvested by centrifugation and the pellet was frozen in liquid nitrogen.

## RNA extraction and analysis

Total RNA was extracted using the hot acid phenol protocol (*Collart and Oliviero, 2001*). Poly(A)$^+$-RNA were obtained by two successive rounds of purification using oligo (dT)$_{25}$ magnetic beads (New England Biolabs, Ipswich, MA) following the manufacturer's protocol. Northern blots were carried out on poly(A)$^+$-RNA as described in *Neil et al. (2009)* using $^{32}$P-labeled riboprobes except for *ACT1* for which a $^{32}$P-labeled oligonucleotide was used.

## Library preparation

Approximately 500 ng of poly(A)$^+$-RNA was mixed with 10 units of Antarctic phosphatase (New England Biolabs) in a final volume of 50 µl. After 1 hr at 37°C, the reaction was treated with phenol/chloroform and ethanol precipitated. The RNA pellet was resuspended in 44 µl of water and 10 units (1 µl) of TAP (Epicentre, Madison, WI) and 5 µl of 10× TAP buffer was added. The reaction was incubated 1 hr at 37°C followed by phenol/chloroform extraction and ethanol precipitation. The RNA pellet was resuspended in 5 µl of water. Next, the RNAs were ligated overnight at 16°C with 50 pmoles of the biotinylated oligonucleotide 3041 (*Table 3*) in a 20 µl reaction containing 10 units (1 µl) of T4 RNA ligase I (New England Biolabs) and ATP at a final concentration of 1 mM. RNAs were subsequently fragmented by incubation for 10 min at 70°C after addition of 5 µl of a 50 mM ZnCl$_2$, 50 mM Tris-HCl pH7.4 solution. The reaction was stopped by the addition of 1 µl EDTA 0.5 M and biotinylated RNA were purified using streptavidin magnetic beads according to the manufacturer's protocol (Dynabeads, MyOne streptavidin C1, Life Technologies, Carlsbad, CA). After washing, the beads were resuspended in 20 µl of water and the bound RNAs eluted by incubation for 5 min at 90°C. This fraction is enriched for 5′-ends of capped RNA molecules. Note that the supernatant of the first step of the purification procedure containing RNA fragments corresponding to the body and the 3′-end of the gene not attached to the biotinylated oligonucleotide can be recovered and used to prepare independent libraries. This RNA population can be further fractionated using oligo (dT)$_{25}$ magnetic beads to enrich for 3′-end of RNA molecules. The ~18.5 µl eluate from the streptavidin beads was mixed with 50 pmoles of oligonucleotide 3038 (see *Table 3*), heat denatured for 5 min at 70°C and slowly cooled down to 30°C in a Biorad iCycler PCR machine. Once the temperature had reached 30°C, 6 µl of 5× RT buffer, 1.5 µl of a 10 mM dNTPs solution, 1.5 µl of RNasin (Promega), 180 ng of actinomycin D, 300 units of RevertAid reverse transcriptase (Thermo Scientific, Waltham, MA), and water qsp 30 µl were added. The reaction was incubated 10 min at 30°C, followed by 40 min at

**Table 2**. Yeast strains used in this study

| Strain | Genotype | Reference |
| --- | --- | --- |
| BY4741 | Mat a, *his3Δ1, ura3Δ0, leu2Δ0, met15Δ0* | (*Brachmann et al., 1998*) |
| LMA1057/3401 | BY4741 *can1Δ* | This study |
| LMA1774/2759 | BY4741 *can1Δ, upf1Δ::HIS3MX6* | This study |
| LMA1676/3405 | BY4741 *can1Δ, rrp6Δ::hphMX6* | This study |
| LMA1772 | BY4741 *can1Δ, upf1Δ::KANMX6, rrp6Δ::HPHMX6* | This study |
| LMA1790 | BY4741 *can1Δ, upf1Δ::KANMX6* | This study |
| LMA2758 | BY4741 *can1Δ* | This study |
| LMA2760 | BY4741 *can1Δ, xrn1Δ::KANMX6* | This study |
| LMA2762 | BY4741 *can1Δ, xrn1Δ::KANMX6, upf1Δ::HIS3MX6* | This study |
| LMA2921/3403 | BY4741 *can1Δ, set2Δ::KANMX6* | This study |
| LMA2922 | BY4741 *can1Δ, set2Δ::KANMX6, upf1Δ::HIS3MX6* | This study |
| LMA3409 | BY4741 *can1Δ, upf1Δ::HIS3MX6, rrp6Δ::HPHMX6* | This study |

**Table 3.** Oligonucleotides used in this study

| Name | Sequence 5′-3′ | |
|---|---|---|
| ACT1-1407− | ACACTTGTGGTGAACGATAGATGG | P³² labelled probe |
| YMR114c-839+ | ATCGAGGTGTAAAGGGTG | Synthesis of probe |
| T7-YMR114C-1068−* | **TAATACGACTCACTATAGGG**CCTCTGGAGTCTTTCTGG | Synthesis of probe |
| YIL136w-1013+ | ACTGGTGGTCTGGATGG | Synthesis of probe |
| T7-YIL136w(+)115−* | **TAATACGACTCACTATAGGG**TGCCACTAATTTACTCCG | Synthesis of probe |
| NEL025c-35+ | AACAAATGCCAAGTCGGGAC | Synthesis of probe |
| T7-NEL025c-263−* | **TAATACGACTCACTATAGGG**AAACGTTTGGTAAGAACTC | Synthesis of probe |
| SUT093_fwd | GAGTCCAGCGTCTCTACAC | Synthesis of probe |
| T7-SUT093_rev* | **TAATACGACTCACTATAGGG**GACTTAATTGTCGTTGCTAGGAC | Synthesis of probe |
| SUT338_fwd | GAAAGACCGAAGGTGAAGAG | Synthesis of probe |
| T7-SUT338_rev* | **TAATACGACTCACTATAGGGG**TGGTACAGCCCTGTGTTCC | Synthesis of probe |
| SUT779_fwd | AACGAGGGAACTAGCCAG | Synthesis of probe |
| T7-SUT779_rev* | **TAATACGACTCACTATAGGG**CTCTTCATCATCTGTGGAG | Synthesis of probe |
| TPO2(+)131− | GTATGTAGAAATGTCCGACG | Synthesis of probe |
| T7-TPO2-1798+* | **TAATACGACTCACTATAGGG**GTAAGGGCTTGAGAC | Synthesis of probe |
| MAL12/32-1723− | GATTCTACCTTCCCATGG | Synthesis of probe |
| T7-MAL12/32-1161+* | **TAATACGACTCACTATAGGG**TCAAGGTCAGGAGATAGG | Synthesis of probe |
| XUT3F5-fwd | AGGAAAATGGGACTACAG | Synthesis of probe |
| T7-XUT3F5-rev* | **TAATACGACTCACTATAGGG**TGTAAAAGGGCACAGTC | Synthesis of probe |
| 3041† | 5BioTEG/CTTTCCCTACACGACGCTCTTCCGATCT**NNNNCGCGrCrGrNrN** | Ligation with TAP treated RNA |
| 3118† | 5BioTEG/CTTTCCCTACACGACGCTCTTCCGATCT**NNNNGCCGrCrGrNrN** | Ligation with fragmented RNA |
| 3038† | GTTCAGACGTGTGCTCTTCCGATCTNNNNNNN | Reverse transcription |

*The sequence in bold face corresponds to the T7 promoter sequence.
†The sequence in bold face corresponds to the tag used to identify the 5′ end of the cDNAs. r stands for ribonucleotide.

42°C, 10 min at 55°C, 10 min at 60°C and 15 min at 75°C. RNAs were then degraded by incubation for 10 min at 75°C after the addition of 3 µl of 1 N NaOH. The reaction was quenched by the addition of 3 µl of 1 N HCl and precipitated by the addition of 3 volumes of 100% ethanol and 0.1 volume of 3 M sodium acetate pH 5.2. The precipitated cDNA was subjected to 5 cycles of PCR amplification in 20 µl with Phusion DNA polymerase (Thermo Scientific) using Illumina (San Diego, CA) multiplexing PCR primer 1.0 and 2.0 followed by an additional 6 to 8 cycles of amplification using Illumina multiplexing PCR primer 1.0 and one PCR primer index. The reaction was purified with Agencourt AmPure XP beads (Beckman Coulter, Indianapolis, IN) following manufacturer's instructions at a 1.8 × concentration and eluted in 20 µl water. Libraries were quantified using Qubit (Life Technologies) and sequenced on an Illumina HiSeq 2000 or 2500 with 50 or 100 bases single-end reads. To validate our method for the identification of TSSs, we compared libraries made with or without treatment of the RNAs with TAP. 500 ng of poly(A)⁺-RNA obtained by purification with oligo (dT)₂₅ magnetic beads starting from *S. cerevisiae* or *S. pombe* (used as an internal reference for normalization) were dephosphorylated using Antarctic phosphatase as described. Then, *S. pombe* RNAs and one half of the RNAs extracted from *S. cerevisiae* were treated with TAP. The second half of *S. cerevisiae* RNAs was mock treated. Before ligation with the biotinylated oligonucleotide 3041, each half of *S. cerevisiae* RNAs was mixed with an equivalent quantity of calf intestinal alkaline phosphatase (CIP)/TAP-treated RNAs from *S. pombe*. The subsequent steps of the library preparation were identical to the ones described above.

For analysis of the *xrn1Δ* mutant compared with the *upf1Δ* mutant, since the absence of Xrn1 leads to the accumulation of decapped RNAs (*Hsu and Stevens, 1993*), we had to modify the protocol used for library preparation. Instead of the CIP/TAP treatment before ligation with the biotinylated oligonucleotide 3118, RNA was fragmented with $ZnCl_2$ and subsequently phosphorylated with T4 polynucleotide kinase (Thermo scientific). The subsequent steps of library preparation were identical to the ones described above for the 5′-end. Moreover, since the absence of Xrn1 affects the overall mRNA content (*Sun et al., 2013*), an aliquot of a *S. pombe* culture was added to the cell pellet before RNA extraction for library preparation to provide an independent internal control for normalization.

## Data analyses

### Illumina reads treatments

Duplicated reads, identified using the random sequence tags within the ligated oligonucleotides (oligonucleotides 3041 and 3118 in *Table 3*) were first filtered out. Then reads corresponding to the 5′-ends of cDNA fragments were extracted using the tag present in the added oligonucleotide (see *Table 3*) for all the unique reads. After removal of the tag, the resulting reads were mapped using bowtie (version 2.2.3 with the following parameters: –N 1 –p 1 —no-unal –D 15 –R 2 –L 22 –I S,1,1.15) and a compilation of *S. cerevisiae* genome (S288C reference sequence, Release 64 obtained from the Saccharomyces Genome Database (SGD) [http://www.yeastgenome.org/]) and *S. pombe* genome (ASM294 reference sequence, v2.19 obtained from PomBase [http://www.pombase.org/]) as reference genomes.

### Mapped reads processing

For libraries L5p_01 to L5p_10 (see *Table 1*) used for TSS sequencing, the 5′-end positions of the resulting mapped reads were used as TSS positions and extracted to wig files. For samples of the libraries LT_01 and LT_02, used for RNAseq, reads corresponding to the whole transcripts and full read coverage were extracted to wig files.

### Peak calling

Libraries L5p_01 to L5p_03 and L5p_04 to L5p_06 were regrouped into clusters (TSSCs) using the peak calling method described in *Neil et al. (2009)*. Parameters were optimized by maximizing the number of ORFs assigned to only one TSSC while minimizing the number of ORFs not assigned to any TSSC. As parameters, we finally chose a threshold of four reads in at least one of the samples used for the clustering and a maximum distance between two consecutive TSSs within a cluster of 50 nucleotides. For library L5p_01 to L5p_03, TSSCs were generated with the code TSSC_upf1_rrp6 (*Supplementary file 1*). For library L5p_04 to L5p_06, TSSCs were generated with the code TSSC_upf1_set2 (*Supplementary file 3*).

### Differential expression

TSSCs and transcript differential expression were calculated using DESeq2 (*Love et al., 2014*) within the SARTools pipeline (https://github.com/PF2-pasteur-fr/SARTools). For TSSCs, the mean TSS read counts in the region 50 to 10 nucleotides upstream of the ATG of each non-dubious ORF were used as an internal standard for size factor determination. For transcripts, the *S. pombe* transcripts read counts were used as the internal standard for size factor determination.

### Identification of TSSCs

TSSCs were assigned to annotations and sorted in a defined class according to their relative positions to the linked features. TSSCs overlapping the 5′-UTR mRNA sequences in the sense orientation were assigned to mRNAs (O-TSSCs in *Supplementary files 1, 3*). TSSCs mapped within the ORF or the 3′-UTR sequences and in the sense orientation were called B-TSSCs. TSSCs overlapping an mRNA sequence but in the antisense orientation were called A-TSSCs. C, S, X, and F TSSCs (*Supplementary files 1, 3*) correspond to TSSCs overlapping in the sense orientation with the 5′-end or found within CUTs, SUTs, XUTs and stable ncRNAs and transposable elements, respectively. All other TSSCs were classified as intergenic (I in *Supplementary files 1, 3*). CUTs, XUTs, and SUTs constituting overlapping transcript populations, when a TSSC was assigned to a pervasive transcript annotated in more than one of these classes we arbitrarily associated the corresponding TSSC in priority to CUTs, then to XUTs and finally to SUTs. mRNA coordinates were extracted from data of *Pelechano et al. (2013)*. ORFs, tRNAs, rRNAs, small nuclear RNAs (snRNAs), and sn(o)RNAs coordinates were retrieved from the SGD (http://www.yeastgenome.org/). CUT and SUT coordinates were retrieved from *Xu et al. (2009)*, while XUT

coordinates were from *van Dijk et al. (2011)*. We also used the CUTs described in *Neil et al. (2009)*. ORF ATG coordinates were corrected for 150 of them identified as misannotated by *Park et al. (2014)*.

## Filtering of TSSCs

False positive TSSCs were filtered out using two criteria: the signal to noise ratio (SNR) and TSS accuracy.

SNR was defined as the log2 of the ratios of read counts of TAP samples vs no-TAP samples. For TSSC_upf1_rrp6 we used libraries L5p_07 and L5p_08 and for TSSC_upf1_set2 we used libraries L5p_09 and L5p_10. We choose a SNR limit of 0.55.

TSS accuracy was defined as the proportion of TSSs within a TSSC mapped on a purine preceded by a pyrimidine vs the total count of TSSs mapped within the TSSC. We choose a minimum TSS accuracy of 80%.

## Distribution of individual TSSs around start codons

Individual TSS counts and read counts were used to determine the repartition of the TSSs around the ATGs of annotated ORFs (−200 to +100 nucleotides around ORFs ATGs). We used the ORF ATG positions retrieved from the SGD and corrected as described above in the 'Identification of TSSCs' section.

## Identification of ORFs following individual TSSs

For each TSS identified in the region from −200 to +100 nucleotides around ORF ATGs, we looked for the two subsequent ATGs. When the TSS was upstream of the annotated ATG start codon, we determined whether or not it was the first encountered one. When the TSS was within the ORF, we determined whether the ATGs were in the same phase as the natural ORF or not.

## Determination of ORF and 3′-UTR sizes for the different features

For mRNAs, ORF start, ORF end and 3′-end coordinates obtained, as described above in the 'Identification of TSSCs' section, were used to calculate the sizes of the ORF and the 3′-UTR. For XUTs and SUTs, ORF start was determined as the coordinate of the first ATG after the annotated start obtained in *Xu et al. (2009)* and *van Dijk et al. (2011)*. ORF ends were determined as the coordinate of the first in phase stop codon encountered after the previously selected ATG start codon. The region between the end of this potential ORF and the end of the XUT or SUT was considered to be the 3′UTR.

## Nucleotide use

We determined the proportion of A, T, G, and C at each position in the region spanning −200 to +100 nucleotides around the ATGs of ORFs according to the annotation found in the SGD and corrected as described above.

## Codons and amino acids use

We determined the number of ATGs within the first 1000 nucleotides after the ORF start codon based on the information available in the SGD. To avoid the overrepresented value of the start ATG codon, the first three nucleotides were not included in the analysis. The ratios between phased ATG and total ATG counts were computed in a window of nine nucleotides. The same method was applied for all other codons. To estimate the differential use of a codon between the start and the body of the ORFs, we normalized the value for each codon taking into account the mean ratio for this particular codon in the region comprised between 500 and 1000 nucleotides after the start codon.

We used the protein sequences retrieved from SGD and omitted the first methionine of the proteins.

## Transcripts expression levels

We used the coverage of LT_01 and LT_02 libraries to calculate transcript expression levels of mRNAs, SUTs and XUTs. The coordinates of the different types of transcripts were obtained as described above. We calculated the expression levels for the four samples of both libraries and then used DESeq2 for normalization, estimation of fold change and mean signal calculation (*Supplementary file 2*).

## Mean NFR density

The mean NFR density has been calculated using a nucleosome definition in *Kaplan et al. (2009)*. We collected the nucleosome signal in regions from −1000 to +1000 nucleotides around each TSS, aligned on TSS position and calculated means for each position.

## Accession number

The data reported here have been deposited in NCBI GEO under the accession number GSE64139.

## Acknowledgements

We thank D Libri, B Séraphin and M Fromont-Racine for discussions and for critical reading of the manuscript.

## Additional information

### Funding

| Funder | Grant reference | Author |
| --- | --- | --- |
| Institut Pasteur | | Christophe Malabat |
| Centre National de la Recherche Scientifique | | Alain Jacquier |
| Agence Nationale de la Recherche | ANR-2011-BSV6-011-02 | Alain Jacquier |
| Agence Nationale de la Recherche | ANR-14-CE-10-0014-01 | Alain Jacquier |

The funders had no role in study design, data collection and interpretation, or the decision to submit the work for publication.

### Author contributions

CM, CS, Analysis and interpretation of data, Drafting or revising the article; FF, Conception and design, Acquisition of data, Analysis and interpretation of data, Drafting or revising the article; LM, Acquisition of data, Analysis and interpretation of data; AJ, Conception and design, Analysis and interpretation of data, Drafting or revising the article

## Additional files

### Supplementary files

• Supplementary file 1. TSS clusters identified in WT, *upf1Δ*, *rrp6Δ* and *upf1Δrrp6Δ* cells.

• Supplementary file 2. Reads count for whole transcriptome analysis in wild type, *xrn1Δ*, *upf1Δ* and *upf1Δxrn1Δ* cells.

• Supplementary file 3. TSS clusters identified in WT, *upf1Δ*, *set2Δ* and *set2Δupf1Δ* cells.

• Supplementary file 4. List of genes with identified iTSSs and/or uORFs.

### Major datasets

The following dataset was generated:

| Author(s) | Year | Dataset title | Dataset ID and/or URL | Database, license, and accessibility information |
| --- | --- | --- | --- | --- |
| Feuerbach F, Malabat C, Ma L, Saveanu C, Jacquier A | 2015 | Quality control of transcription start site selection by nonsense-mediated-mRNA Decay | http://www.ncbi.nlm.nih.gov/geo/query/acc.cgi?acc=GSE64139 | Publicly available at NCBI Gene Expression Omnibus (GSE64139). |

The following previously published datasets were used:

| Author(s) | Year | Dataset title | Dataset ID and/or URL | Database, license, and accessibility information |
| --- | --- | --- | --- | --- |
| Xu Z, Wei W, Gagneur J, Perocchi F, Clauder-Münster S, Camblong J, Guffanti E, Stutz F, Huber W, Steinmetz LM | 2009 | Transcription profiling of wild type yeast grown with ethanol, glucose and galactose and the deletion mutant of Rrp6 to identify transcription start and end positions | http://www.ebi.ac.uk/arrayexpress/experiments/E-TABM-590/ | Publicly available at ArrayExpress (E-TABM-590). |

| Author(s) | Year | Dataset title | Dataset ID and/or URL | Database, license, and accessibility information |
|---|---|---|---|---|
| Arribere JA, Gilbert WV | 2013 | Roles for Transcript Leaders in Translation and mRNA Decay Revealed by Transcript Leader Sequencing | http://www.ncbi.nlm.nih.gov/geo/query/acc.cgi?acc=GSE39074 | Publicly available at NCBI Gene Expression Omnibus (GSE39074). |

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
