## [Decision Letter]

Thank you for sending your work entitled “Quality control of transcription start site selection by nonsense-mediated-mRNA decay” for consideration at *eLife*. Your article has been favorably evaluated by James Manley (Senior editor) and two reviewers, one of whom is a member of our Board of Reviewing Editors.

The Reviewing editor and the other reviewer discussed their comments before we reached this decision, and the Reviewing editor has assembled the following comments to help you prepare a revised submission.

Overall the study by the Jacquier laboratory is timely and contains very interesting information adding to our knowledge of how pervasive transcription is suppressed post-transcriptionally by RNA turnover.

Before acceptance the following points should be addressed:

Major points:

1) A technical concern relates to the TSS sequencing method, which relies on the treatment of poly(A)^+^ RNA with phosphatase to prevent degradation intermediates with a 5' phosphate from ligating to the biotinylated adaptor oligo. Since transcripts with 5'ends originating downstream of their annotated TSSs (B-TSSC, iTSS) are identified, it seems crucial to control for the efficiency of the phosphatase treatment, as many of these transcripts may otherwise represent degradation intermediates rather than transcripts with alternative 5'ends. Finding the same consensus sequence motif as at canonical TSSs helps, but do the authors have any direct means to control for the effective exclusion of such species from the libraries?

2) Differential expression analysis should be done according to today's best practice, which includes normalization of the data that accounts for differences in sequencing depths between samples (i.e. size factor normalization [see DESeq2] rather than rpkm).

Minor points:

1) The authors' seem somewhat biased towards the idea that every transcript that is repressed by some factor constitutes noise. Is this really always clear?

2) The authors identify a consensus sequence, which basically is identical for all types of RNAPII transcripts: A(N)6PyPu, where the Pu is the TSS. Is it possible to investigate/estimate what fraction of such sites in the genome is functioning as TSSs based on the TSS sequencing data?

3) The term “synergistic”, used in the subsection “Identification of additional pervasive transcripts” (“it had a synergistic effect on some ‘intergenic’…”): could this synergism simply be an additive effect that is hidden by the generally low expression of these RNAs?

4) What is the reason for the overall rather low mapping rate of the sequencing data (sometimes as low as 30%)?

5) Throughout the manuscript only few statistical tests are applied. While in many cases the displayed effects appear convincing, it would be useful to, e.g., evaluate whether the small difference in Figure 7 is statistically significant.

---

## [Author Response]

*Major points*:

*1) A technical concern relates to the TSS sequencing method, which relies on the treatment of poly(A)*^*+*^
*RNA with phosphatase to prevent degradation intermediates with a 5' phosphate from ligating to the biotinylated adaptor oligo. Since transcripts with 5'ends originating downstream of their annotated TSSs (B-TSSC, iTSS) are identified, it seems crucial to control for the efficiency of the phosphatase treatment, as many of these transcripts may otherwise represent degradation intermediates rather than transcripts with alternative 5'ends. Finding the same consensus sequence motif as at canonical TSSs helps, but do the authors have any direct means to control for the effective exclusion of such species from the libraries*?

We agree that, in order to use the mapping of RNA 5’-ends as marks of transcription start sites, it is critical to control that they originated from capped-RNA ends. Even though we have not controlled the efficiency of the phosphatase treatment step specifically, we think that the best control for this is the experiment we performed comparing the results obtained with or without treatment with tobacco acid pyrophosphatase (TAP) shown in Figure 1. The reads from the minus-TAP experiment were used to measure the background of reads originating from 5’-mono phosphate RNAs remaining after phosphatase treatment, which was estimated as being, overall, lower than 2.5% (see end of the Results section “TSS sequencing technique”). The genome-wide data for this experiment, deposited in the GEO database, were used to filter out a few possible false positive TSSCs, as explained in the Materials and methods section “Filtering TSSCs”. Note that, concerning iTSSs, we purposely showed a zoom of the corresponding regions in Figure 1, which clearly demonstrates that the vast majority or reads mapped just after the START codons do originate from capped-RNAs since they are observed only after the TAP treatment. We have now added an appropriate sentence when first describing iTSSs to mention this point (see “Almost half of the coding genes produced transcript isoforms sensitive to NMD”).

*2) Differential expression analysis should be done according to today's best practice, which includes normalization of the data that accounts for differences in sequencing depths between samples (i.e. size factor normalization [see DESeq2] rather than rpkm)*.

We agree with this comment (even though we did not use rpkm for size factor normalization). We have now re-analysed all the data using DESeq2. This allowed more rigorous statistical analyses of the data and all figures and numbers were changed accordingly. In particular, we now use the standard Wald-test provided in DESeq2 with the usual threshold for the p-value of 0.05, instead of a log_2_ threshold of 1, to determine the transcripts, TSSs or TSSCs significantly expressed differentially in the different strains. Note that, although to two approaches are markedly different, the results were very similar and using this more rigorous approach did not changed the overall conclusions drawn.

*Minor points*:

*1) The authors' seem somewhat biased towards the idea that every transcript that is repressed by some factor constitutes noise. Is this really always clear*?

No, we certainly do not consider that all transcripts targeted for degradation by quality-control mechanisms constitute transcriptional noise. For example, in some cases, it has been clearly shown that the transcription of some non-coding RNAs can have a biological function, in particular for gene expression regulation by transcriptional interference. The analysis of such functions for the thousands pervasive transcripts identified in this work is out of the scope of this manuscript. Also, and as discussed in the manuscript, a substantial fraction of mRNAs is down regulated by NMD and it certainly does not correspond to transcriptional noise. The different use of TSSs has also previously been shown to be a mean of regulation. We have now added a short paragraph in the Discussion to mention this possibility (see “These two observations reveal an important role for NMD […] some genes of the nucleotide biosynthetic pathway (27; 44)”).

*2) The authors identify a consensus sequence, which basically is identical for all types of RNAPII transcripts: A(N)6PyPu, where the Pu is the TSS. Is it possible to investigate/estimate what fraction of such sites in the genome is functioning as TSSs based on the TSS sequencing data*?

We identified 1,775,474 sites in the genome with the exact consensus A(N)6PyPu and 85,714 (4.8%) were used as transcription start sites. A sentence was added in the Discussion (“Importantly, this consensus sequence […] different classes of transcripts identified in this study”).

*3) The term “synergistic”, used in the subsection “Identification of additional pervasive transcripts” (“it had a synergistic effect on some ‘intergenic’…”): could this synergism simply be an additive effect that is hidden by the generally low expression of these RNAs*?

Indeed, for a number of the pervasive transcripts, this can be true. Yet, for some of them, exemplified by the novel pervasive transcripts found antisense to MAL12/32 and shown in Figure 4, there is a strong signal in the double mutant and no detected signal in the single mutants. We think that the most likely explanation for this observation is, indeed, that the effect of the two mutations is synergistic and not only additive. We have modified the corresponding sentence in the Results section to attenuate the statement and to more clearly point to the above-mentioned example (“Furthermore, the absence of both Rrp6 and Upf1 […] *upf1∆rrp6∆* double mutant (see for example the transcript 268 found antisense to MAL12/32 in Figure 4”).

*4) What is the reason for the overall rather low mapping rate of the sequencing data (sometimes as low as 30%)*?

Before mapping there is a filtering step, as it is described in materials and methods. This step first collapse reads with the same sequence and the same 5’-end random tag to eliminate PCR duplicates. We added this information in Table1 in the revised version of the manuscript, to indicate the reads count after collapsing the reads. Duplication level can be high for some samples and sometimes only 30 % of the reads are really unique. On the contrary, mapping level, considering unique reads count as initial count, is around 95% for all the samples.

*5) Throughout the manuscript only few statistical tests are applied. While in many cases the displayed effects appear convincing, it would be useful to, e.g., evaluate whether the small difference in*
Figure 7
*is statistically significant*.

The use of the three biological replicates in DESeq2 allowed a number of statistical analyses to be performed. For example, the number of differentially expressed transcripts are not defined by a threshold anymore but by the Wald-test provided in DESeq2. For Figure 7 (now 7C), we provided the requested p-value (ANOVA test) in the figure. Moreover, we added an additional panel (now 7B) for the statistical analysis of Figure 7.